# Standardized metrics can reveal region-specific opportunities in community engagement to aid recruitment in HIV prevention trials

Gail B. Broder[1]*, Jonathan P. Lucas[2], Jontraye Davis[2], Stephaun E. Wallace[1], Nandisile Luthuli[1], Kagisho Baepanye[1], Rhonda R. White[2], Marcus Bolton[2], Cheryl Blanchette[2], Michele P. Andrasik[1]

1 Vaccine and Infectious Disease Division, Fred Hutchinson Cancer Research Center, Seattle, Washington, United States of America, 2 Science Facilitation Department, FHI 360, Durham, North Carolina, United States of America

* gbroder@fredhutch.org

**Data Availability Statement:** All relevant data are within the manuscript.

## Abstract

Good Participatory Practice (GPP) guidelines support and direct community engagement practices in biomedical HIV prevention trials, however no standardized metrics define the implementation and evaluation of these practices. Collaboratively, the Community Program staff of the HIV Vaccine Trials Network (HVTN) and the HIV Prevention Trials Network (HPTN) created a metric to describe, monitor, and evaluate one component of GPP, recruitment practices, in two HIV monoclonal Antibody Mediated Prevention (AMP) clinical trials, HVTN 703/HPTN 081 and HVTN 704/HPTN 085. Through consultation with community representatives from each clinical research site (hereafter "site(s)"), who made up the study Community Working Groups, recruitment strategy descriptors were developed for both trials to characterize responses to "How did you hear about the AMP study?" The Community Working Groups also helped to define and establish time points that were selected to allow comparisons across sites. Data were collected by 43 of 46 clinical research sites from January 1, 2017 to February 28, 2018. All 43 sites used multiple recruitment strategies successfully, but strategies varied by region. Globally, referrals was the most efficient and effective recruitment strategy as evidenced by the screening: enrollment ratio of 2.2:1 in Africa, and 2.1:1 in the Americas/Switzerland. Print materials were also valuable globally (3:1 Africa, 4.2:1 Americas/Switzerland). In Africa, in-person outreach was also quite effective (2.3:1) and led to the most enrollments (748 of 1186, 63%). In the Americas/Switzerland, outreach was also effective (2.6:1), but internet use resulted in the most screens (1893 of 4275, 44%) and enrollments (677 of 1531, 44%), compared to 12 of 2887 (0.4%) and 2 of 1204 (0.1%) in Africa, respectively. Standardized metrics and data collection aid meaningful comparisons of optimal community engagement methods for trial enrollment. Internet strategies had better success in the Americas/Switzerland than in sub-Saharan African countries. Data are essential in outreach staff efforts to improve screening-to-enrollment ratios. Because the effectiveness of recruitment strategies varies by region, it is critical that clinical research

**Funding:** The authors received no specific funding for this work.

**Competing interests:** The authors have declared that no competing interests exist.

sites tailor community engagement and recruitment strategies to their local environment, and that they are supported with resources enabling use of a range of approaches.

## Introduction

Community engagement is a key component of biomedical HIV prevention research [1]. It is a process of demonstrating mutual respect between community stakeholders, researchers, research sponsors, and other stakeholder groups to facilitate inclusion and collaboration with communities where clinical trials will be conducted [2]. The community engagement process continues throughout the lifecycle of a clinical trial, from concept generation through results dissemination and study close-out [3, 4]. Community engagement is inclusive of community education, study recruitment, retention, and product adherence, and is recognized as a significant contributor to these important clinical trial elements [2, 5].

Research networks funded by the National Institutes of Allergy and Infectious Diseases (NIAID) Division of AIDS (DAIDS) include community engagement programs as a core component of network operations [6]. The HIV Vaccine Trials Network (HVTN) and the HIV Prevention Trials Network (HPTN), funded through NIAID, are international collaborations of scientists, clinical trial sites, and community representatives who work with governments and industry in the global search for safe and effective ways to prevent HIV. The HVTN is focused on finding a preventive HIV vaccine. The HPTN's research scope includes using antiretroviral medications for HIV prevention, and novel social and behavioral interventions. The HVTN and the HPTN have come together to conceptualize, develop, and conduct two efficacy studies using a novel HIV-1 broadly neutralizing antibody, VRC01. HVTN 703/HPTN 081 (ClinicalTrials.gov NCT02716675) is being conducted in sub-Saharan Africa with heterosexual women, and HVTN 704/HPTN 085 (ClinicalTrials.gov NCT02568215) is being conducted in the United States, Brazil, Peru, and Switzerland with cisgender men and transgender individuals who have sex with men. The two parallel trials are also known as the Antibody Mediated Prevention (AMP) studies.

The AMP studies are the first efficacy trials testing VRC01, a broadly neutralizing antibody that binds to the CD4 binding site on the HIV-1 envelope, thus blocking HIV from binding to and infecting human T cells. In lab and animal tests, this antibody was shown to neutralize HIV in approximately 90% of tested global circulating HIV-1 isolates [7].

The two ongoing AMP studies are double-blind, randomized, placebo-controlled trials. HVTN 704/HPTN 085 has fully enrolled 2701 cisgender men, transgender men, and transgender women who have sex with men; HVTN 703/HPTN 081 has fully enrolled 1924 heterosexual women. Study participants were assigned to one of three study regimens (active lower dose, active higher dose, or placebo) and were followed for 26 months. The study product is administered via intravenous infusions every eight weeks, for a total of 10 infusions per participant during the trial. The studies are designed to assess whether the VRC01 antibody is safe and well-tolerated; whether VRC01 reduces acquisition of HIV; and, if reduced HIV acquisition is observed with VRC01, what concentration and activity of VRC01 is associated with that protection [8].

The Network Community Engagement Program staff of the HVTN and HPTN, based in the leadership and operations centers, facilitated broad community engagement in the conceptualization and conduct of the AMP studies, utilizing the Good Participatory Practice (GPP) Guidelines for Biomedical HIV Prevention Trials [3, page 13]. The GPP guidelines were

developed by UNAIDS and AVAC to standardize practices globally for stakeholder engagement in biomedical HIV prevention trials [3, 9]. *"The GPP guidelines set global standard practices for stakeholder engagement. When applied during the entire lifecycle of a biomedical HIV prevention trial, they enhance both the quality and outcomes of research. While there is much guidance in the field on how to conduct trials, the GPP guidelines are the only set of global guidelines that directly address how to engage stakeholders in the design, conduct, and outcome of biomedical HIV prevention trials [3, page 68]."* While GPP is utilized for research conducted by the HVTN and HPTN, it is not without challenges. These include the representativeness of community members for the broader community, and impartiality of advisory community structures funded by clinical research sites [4].

The Networks support partnerships between communities and researchers throughout the entire research process and engage with communities at the research sites where studies are conducted as well as with other key internal and external stakeholders through regular tele-/videoconferences, protocol team meetings, site assessment visits, study-specific consultations and trainings, and workshops. External stakeholders include advocacy groups and civil society groups such as pastors, traditional healers, tribal chiefs, Ministry/Department of Health representatives, community-based organizations, and others. Although this community representation is in place and fully utilized, the GPP guidelines do not provide specific metrics for measuring or assessing the implementation of community engagement. This project aimed to characterize the recruitment methods used and to identify how these methods worked across the two AMP efficacy studies. It was with this in mind that we focused on "engaging stakeholders in the conduct of research" as one element of GPP to measure. The members of communities who are at risk for HIV acquisition are stakeholders in study recruitment efforts, as are the members of our community advisory bodies who provided input into how recruitment would be conducted.

Network Community Engagement Program staff in the HPTN and HVTN are permanent teams of full-time staff in place to provide technical assistance, capacity building, and leadership to community engagement efforts at the local sites across the networks. Network Community Engagement Program staff lead and facilitate implementation of HVTN's and HPTN's community engagement approach. Our approach relies on community advisory structures such as Community Working Groups and site Community Advisory Boards (CABs) (Fig 1).

The purpose of the AMP Community Working Groups is to ensure that the principles of community involvement are the foundation of all community engagement activities at each clinical research site, and to facilitate community participation throughout the research process (concept development, study implementation, and results dissemination), corresponding to the GPP principle of involving the community in study conduct quoted above. The AMP Community Working Groups are composed of one community educator and one community advisory board representative from each participating site. They share information on successes and challenges related to community engagement, recruitment, and retention from across the spectrum of participating sites. These groups also provide guidance to the study protocol teams. Goals of the AMP Community Working Groups include providing input into protocol development; adapting sample consent forms for local use and development of other study-related materials; increasing capacity of members through participation in protocol-specific trainings and regional workshops; informing strategies for recruitment and retention; and assisting in monitoring any emerging issues in the community. Members of the AMP Community Working Groups participate in regular conference calls, face-to-face meetings, and workshops. Protocol leadership does not participate on AMP Community Working Group conference calls in an effort to ensure that community educators and CAB representatives can freely express any concerns related to study conduct without fear of repercussions.

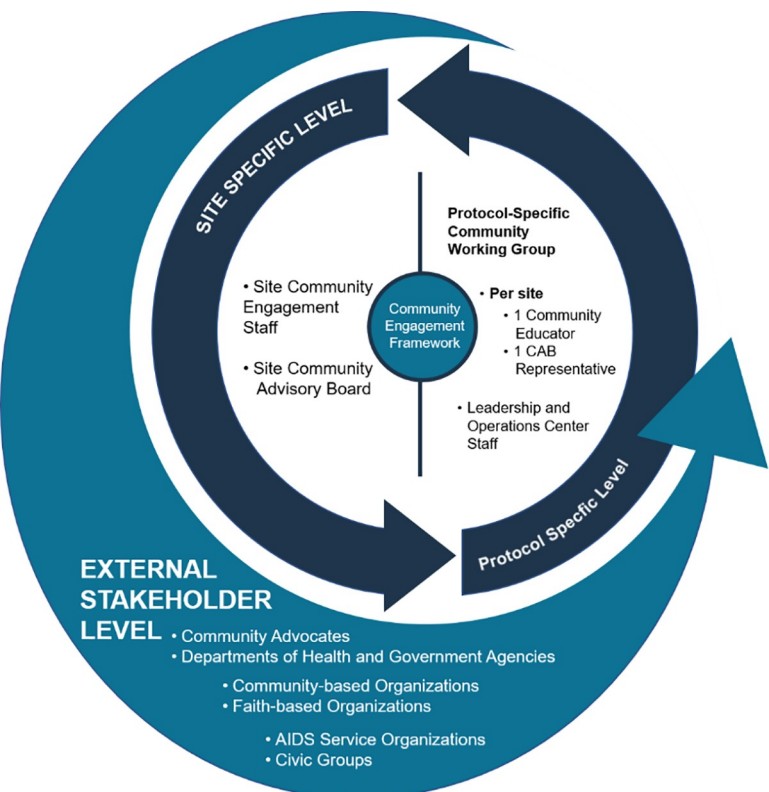

**Fig 1. HVTN/HPTN community engagement framework.**

Clinical research site CABs serve as the voice for the community and study participants in a particular locale [4, 10]. CABs bring specific, unique expertise to the research process by participating in defining the scientific agenda and informing researchers of local issues or concerns that can affect the conduct and successful implementation in that locale [4, 10]. Each clinical research site participating in the AMP studies supports the work of an active CAB to represent the community of potential and enrolled participants by raising research-related issues or concerns that may impact the local participants, local community, or study overall [4, 10].

Community educators are responsible for the development and implementation of site-specific community engagement work plans that outline goals, objectives, and the local scope of work based on a local community needs assessment. Community educators are employed locally by the clinical research sites and are typically full-time employees. They work in collaboration with their local CAB representatives to assess and identify appropriate educational strategies/materials that need to be developed to educate their communities about the research agenda [4, 10]. The networks require that local CABs review these plans and provide input into their development, and the names of CAB members who participate in this review are submitted as contributors to the plan. Community educators are also responsible for assuring that CAB representatives have input into study-specific issues such as addressing community misconceptions, determining appropriate and non-coercive incentives for trial participation and to support retention, as well as determining the package of services that make up the local standard of HIV prevention [10].

The HVTN 703/HPTN 081 Community Working Group is comprised of 40 community representatives and the HVTN 704/HPTN 085 Community Working Group is comprised of

52 community representatives. Both groups are supported by nine Network Community Engagement Program staff from the HVTN and HPTN. In addition, community representatives serve on network committees and protocol teams, providing input to Network efforts well in advance of protocol implementation, such as reviewing draft concepts at the earliest stages of protocol development and reviewing the draft protocol to determine whether it should move forward to implementation.

Our underlying assumption was that community members have the best insight regarding which recruitment strategies would work best in their local cultural and geographic context [10]. Human and financial resources for recruitment are limited, thus assessing the performance and efficiency of recruitment strategies was important to ensure that those resources were maximized to their greatest potential. This framework guided the decision to assess the efficiency of recruitment strategies.

## Methods

Community representatives contributed to the development of local recruitment plans and to identifying the recruitment strategies assessed in this project. Network Community Engagement Program staff for the HVTN and HPTN used a list of recruitment strategies developed by a community working group on a prior preventive HIV vaccine study as a starting point. The list was shared with the Community Working Groups of both AMP studies in discussions prior to study implementation, and the members revised the list for accuracy in the present study context.

The African Community Working Group expressed the importance of categorizing referral sources, so the strategy was split into four separate items: participants, providers of counseling and testing services, people who had attended a male involvement program, and other individuals or groups. The Community Working Group for the Americas and Switzerland did not feel that this was necessary, and compiled all referrals as a single recruitment strategy.

Community Working Group members also discussed how to create definitions and time points for evaluation purposes, looking for time points that would be common across sites. Recognizing that sites varied in their screening processes, they determined that a common time point that would allow cross-site comparison was when a volunteer came into the clinic to give their written consent to be screened for study eligibility. This time point was common across all geographic regions for both studies. This became the point at which all sites asked, "How did you hear about the AMP study?" and recorded the answers. Potential participants were encouraged to name all of the ways in which they had heard about the study, and staff were trained to record them all. Sites also tracked how many of those who were screened ultimately enrolled in the trial and submitted these data to the Network Community Engagement Program staff quarterly. The data presented are for the 13-month period January 1, 2017, through February 28, 2018, and represent 43 of the 46 participating sites. The remaining three sites, one in the U.S. and two in sub-Saharan Africa, elected not to share their data for this analysis.

To determine the screening-to-enrollment ratios for assessment of how efficiently each strategy performed, the number of people screened was divided by the number of people enrolled for each recruitment strategy. It was expected that there would be people who screened for the trial who were found ineligible to enroll, or who were eligible but decided not to participate. As a point of reference, sites were funded with an expectation of screening roughly seven to eight people for each person enrolled. These recruitment strategies were all quite efficient and performed above expectations.

**Table 1. Recruitment strategies used with HVTN 704/HPTN 085.**

| Totals for HVTN 704/HPTN 085: Brazil, Peru, Switzerland, and United States January 1, 2017 –February 28, 2018 | | | | |
|---|---|---|---|---|
| Recruitment Strategy | Initiated the Informed Consent Process by Coming into the Clinic | Enrolled | % of Enrollment | Screening- to-Enrollment Ratio |
| Internet (Craigslist, Social Media, dating websites, www. ampstudy.org, etc.) | 1893 | 677 | 44.2% | 2.8:1 |
| Bar/Street/Pride Festival/other face-to-face outreach conducted by recruiters | 1118 | 421 | 27.5% | 2.6:1 |
| Referrals (from another participant, health care provider, CBO, HIV testing center, etc.) | 428 | 211 | 13.8% | 2.1:1 |
| Print advertising and materials (flyers, posters, palm cards, newspaper ad, etc.) | 474 | 112 | 7.3% | 4.2:1 |
| Other (rollover from another trial, Radio/TV ads, unknown, etc.) | 331 | 103 | 6.7% | 3.2:1 |
| Transit advertising | 31 | 7 | 0.5% | 4.4:1 |

## Results

When recruitment strategy performance data for both AMP cohorts are compared, the regional differences are clearly apparent. Globally, sites utilized multiple recruitment strategies successfully, but the strategies varied in effectiveness, as reflected in screening-to-enrollment ratios, by cohort. Use of referrals was most efficient with screening-t- enrollment ratios of 2.1:1 in the Americas/Switzerland (Table 1) and 2.5:1 in Africa (Table 2). Print materials were modestly valuable globally, with screening-to-enrollment ratios of 4.2:1 (Americas/Switzerland) and 3:1 (Africa). In Africa, in-person outreach was the best practice, accounting for 63% of enrollment with a screening-to-enrollment ratio of 2.3:1. In the Americas and Switzerland, outreach was also effective, bringing in 27.5% of enrollments, but internet use resulted in 44.2% of enrollments. Both outreach and the internet were quite efficient at 2.6:1 and 2.8:1, respectively. Internet-based recruitment success differs more than 300-fold between regions, at 44% of total enrollments in the Americas and Switzerland vs. 0.1% in Africa.

While there was some overlap in the top three strategies for each cohort, their associated enrollment varied substantially. Internet, outreach, and referrals were most effective in the Americas and Switzerland. In Africa, outreach and referrals from any source were most

**Table 2. Recruitment strategies used with HVTN 703/HPTN 081.**

| Totals for HVTN 703/HPTN 081: Botswana, Kenya, Malawi, Mozambique, South Africa, Tanzania, Zimbabwe January 1, 2017 –February 28, 2018 | | | | |
|---|---|---|---|---|
| Recruitment Strategy | Initiated the Informed Consent Process by Coming into the Clinic | Enrolled | % of Enrollment | Screening-to-Enrollment Ratio |
| Face-to-face outreach conducted by recruiters (includes sports events, health fairs, liquor stores, community presentations, "hot spots," etc.) | 1763 | 758 | 63.0% | 2.3:1 |
| Referral by another participant (someone enrolled in AMP or another study) | 560 | 229 | 19.0% | 2.5:1 |
| Referrals (from a health care provider, Community Based Organization/Non-Governmental Organization, CAB, peer educators, friend/family member, etc.) | 326 | 145 | 12.0% | 2.2:1 |
| Referral from HIV counseling and testing program or center | 108 | 26 | 2.2% | 4.1:1 |
| Print materials or advertising (posters, post cards, newspaper ads, etc.) | 66 | 22 | 2.0% | 3.0:1 |
| Other | 39 | 15 | 1.1% | 2.4:1 |
| Referral by someone who participated in a male involvement program | 12 | 6 | 0.5% | 2.0:1 |
| Internet: saw the website www.ampstudy.org.za, or through social media | 12 | 2 | 0.1% | 6.0:1 |
| Radio program or advertising | 1 | 1 | 0.08% | 1.0:1 |

effective, followed distantly by print materials. These findings indicate the need for very different work practices by community engagement site staff and different investment of resources for participants enrolled in each region. For example, the scheduling of staff who work in pairs to conduct outreach activities involves different considerations than the scheduling of staff who will work in online environments from an office.

Use of the study websites, www.ampstudy.org and www.ampstudy.org.br, varied by country within the Americas/Switzerland cohort. The Community Working Group members in the U. S. and Brazil decided to use the websites as a recruitment tool, where potential participants could submit their contact information to a site online, in order to be contacted by staff at the site they had selected. The Community Working Group members in Peru elected to use the study website only for educational purposes, without the ability to collect contact information, but they did run advertising on a number of online platforms. (The Peru website is also www. ampstudy.org; the website geolocates the user and presents the website in Spanish, without the contact form.) The Community Working Group members in Switzerland intended to use the website for recruitment, but they enrolled all of their participants using other strategies while the website was still under ethics committee review, so although the website was created in French, it was never ultimately launched. Thus, the number of people reached through internet strategies varied by the site's country location.

Similar to the sites in Peru, the African Community Working Group members decided to only use the study website www.ampstudy.org.za as an educational tool. They perceived that people in their communities would be reluctant to submit personal contact information over the internet, but rather would seek out the website as a source of information to learn about the trial. They also felt that having a website was viewed as a way of authenticating the study and demonstrating its legitimacy. The website was made available in Chichewa, English, Luo, Portuguese, Sesotho, Setswana, Shona, Swahili, and Zulu, which were selected by the Community Working Group as being the most needed languages for members of their communities.

As shown in Fig 2, accrual progressed well above projected expectations throughout the evaluation period, with the exception of January and February 2018. The leap in January 2018 reflected revisions to enrollment projections based on strong enrollment performance, however, during January and February 2018, sites were instructed to significantly slow accrual due to limited product availability resulting from delays in the manufacturing of the broadly neutralizing antibody. No recruitment deficiencies were identified by Network Community Engagement Program staff during these two months; rather, recruitment strategies were used less frequently due to lower enrollment expectations.

## Discussion

Several studies have highlighted the benefits to inclusion of community representatives in the development of strategies for community engagement and study recruitment [4, 11–17]. Recruitment success for biomedical HIV prevention trials that require thousands of individuals deemed at increased risk for acquisition of HIV is optimized by implementation of multifaceted and complementary community engagement and recruitment strategies [18, 19]. Lack of the establishment of meaningful community engagement strategies can hinder rapport building and the development of culturally appropriate community education and recruitment approaches targeted toward the inclusion of marginalized populations [12]. Ongoing monitoring of community engagement and recruitment strategies can inform and improve methods for participant engagement and ensure cultural relevance while eliminating barriers for study participation [11].

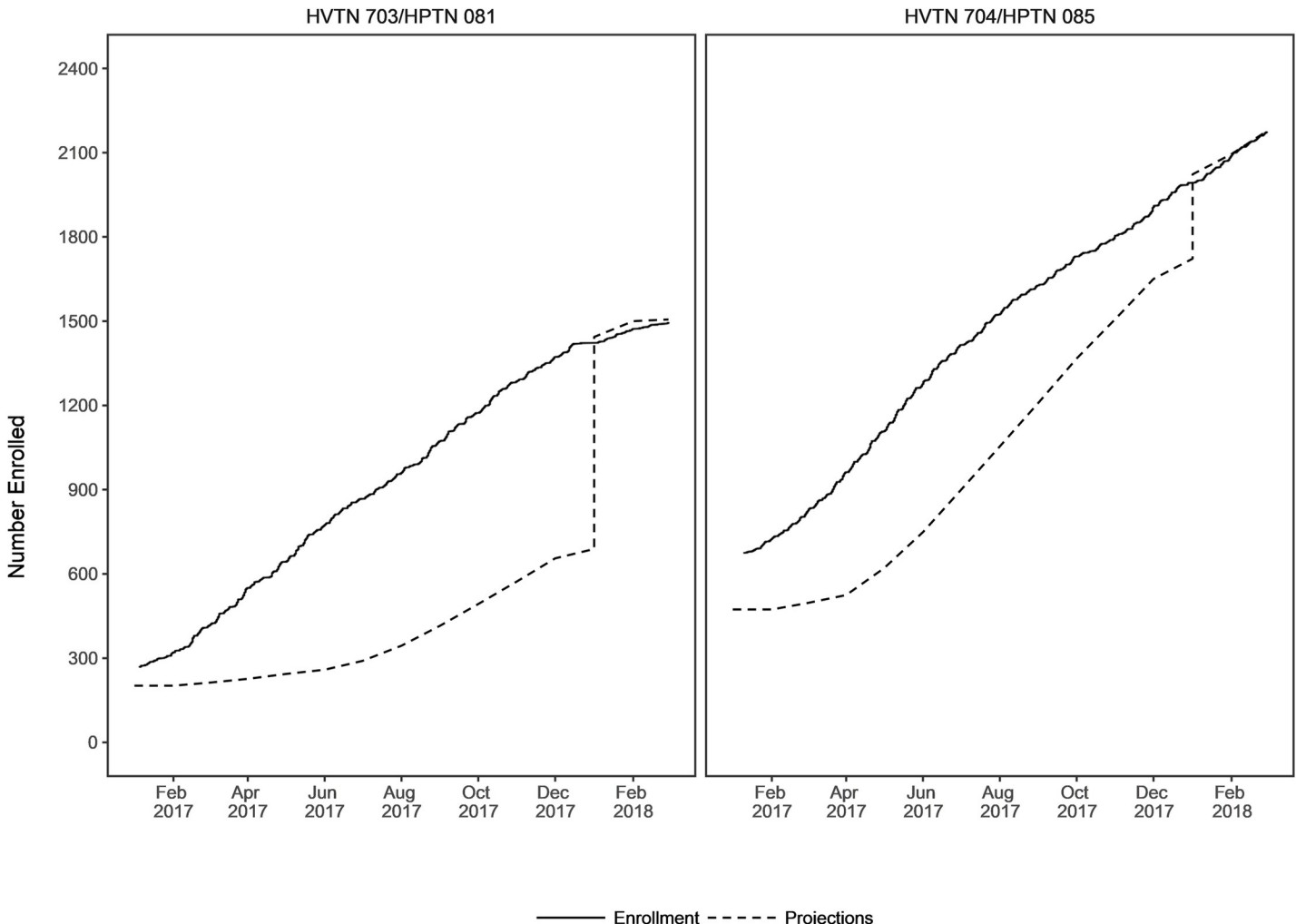

**Fig 2. Projected enrollment and actual enrollment for the AMP studies.**

These data provide a useful starting point for considering how to characterize, monitor, and evaluate community engagement in HIV prevention efficacy trials. The internet and face-to-face outreach were most utilized in the Americas and Switzerland, while face-to-face outreach and referral were most utilized in sub-Saharan Africa. These data demonstrate the importance of using a variety of recruitment strategies, as no single strategy was associated with successful study enrollment [13, 15, 16]. This speaks to the need for investment in robust community engagement resources, both human and financial [11].

These data highlight the importance of consulting with the community to identify recruitment strategies most appropriate for each region [10]. This incorporation of community input supported successful recruitment that was well above projections throughout the accrual period, with the exception of January and February 2018 due to the aforementioned product-dependent, externally imposed slower accrual rate. An example of this can be seen in use of the internet for recruitment in both the Americas/Switzerland and Africa. African community representatives suggested that the internet would not be a useful recruitment tool, but rather would be an educational tool used by potential participants to learn about the study. The AMP Africa website garnered 2,704 page views. In the Americas and Switzerland, the community

suggested that internet recruitment was a vital means of reaching the target populations in places where they already spent time online. The AMP website platforms for the U.S., Brazil, and Peru garnered a total of 66,714 page views. Even when comparing the Peru website and the African website, which were both educational in nature, the page views were 6,199 and 2,704, respectively, further underscoring the difference in web utilization in these regions. Because the study was not designed to determine the role of internet use as a recruitment strategy, we can only describe the associations seen in our data, without suggesting causation.

Use of these varied strategies has implications for models of community engagement staffing and finances. For example, sites using face-to-face outreach successfully may require additional staff members to carry out this work which can occur at nontraditional times outside of normal business hours, while sites relying more heavily on the internet may need fewer staff members working at different times of day, which could include working from home or an internet café rather than from the site office. There are also implications for the hiring practices of community engagement staff so that skill sets, such as the ability to strike up conversations with strangers in public venues, can be assessed. Website development and promotion, including online advertising that drives traffic to those websites throughout the accrual period, are major expenses. In the AMP studies, the costs associated with building the websites and the promotional advertising were handled by the HVTN and HPTN Leadership Operations Centers and not by the sites.

From the Networks' perspectives, these data enabled meaningful comparisons across sites and across regions. Quarterly reviews of site data provided an understanding of how well recruitment strategies were working over time, including identification of "best practices" or "better practices." These strategies were then shared across sites. This sparked new ideas for some sites, enabling sites to learn from and/or duplicate the successes of others. Sites also exchanged strategies that were less effective, perhaps indicating training needs, support needs, or the need to establish new recruitment strategies to replace those not working well. The Network Community Engagement Program staff were able to work with sites individually to support these efforts and provide assistance as needed.

Our study has some limitations. One limitation of this project is knowing whether a participant reported all of the ways they may have heard about the trial, or perhaps only the most recently experienced strategy that was easiest to recall. Social science literature points to the Mere Exposure Effect in which people see or hear a message 10 to 20 times before becoming familiar with it, so the reasons that potential participants gave for how they heard about the study are likely only a limited snapshot of how people may have actually seen or heard about these trials [20]. While study staff were trained to record all answers given by a potential participant, they were not trained in interviewing techniques that may have elicited more detailed responses.

A second limitation is that the data on screening and enrollment was self-reported by each participating research site, and was not validated through any objective methods. A third limitation is the incomplete dataset due to three of the 46 study sites electing not to provide their screening-to- enrollment data. The missing American site enrolled 32 participants, and the two African sites collectively enrolled 134 participants during the period of analysis. Although not part of the dataset, the two African sites represent 10% of the total African enrollment, and the American site represents 2% of the total enrollment in the Americas/Switzerland. Based on Network Community Engagement Program staff experience working with these sites, we do not believe that their data would have been substantively different from the other sites in their respective regions, nor would it have swayed the results substantially.

Another limitation of this study is that the use of the recruitment strategies cannot be broken down by any subpopulations within each region, such as differentiating between cisgender

and transgender persons or between persons of different races and ethnicities. Employing evidence-based community engagement and recruitment strategies supports recruitment of populations deemed vulnerable for acquisition of HIV [21]. Analysis of data for vulnerable populations can identify successful community engagement and recruitment strategies as well as improve cost effectiveness [21]. While our efforts provide an introductory characterization of effective recruitment strategies by geographic region, future studies would benefit from gathering data that can be correlated to the demographics of study participants.

Finally, our study made no attempt to characterize recruitment strategies or harmonize metrics with regard to cultural, geographic, or economic differences that could influence the implementation of the GPP guidelines. This has been noted as a challenge for implementation of GPP in research conducted by Newman et al. [4]. While our consultations with communities did enable us to conduct recruitment that was sensitive to avoiding stigmatizing populations deemed at increased risk for acquisition of HIV, we did not conduct further work to consider the influences of wealth disparities and power imbalances among these populations which could also vary geographically [4, 22]. Future research efforts would benefit from the study of these influences, and how community engagement efforts might address any of the issues that arise from them.

## Conclusions

Use of standardized metrics to assess community engagement practices revealed regional differences in effective ways of recruiting study participants. Low screening-to-enrollment ratios indicate effective use of staff time and available resources. These findings also highlight the wide breadth of strategies needed to effectively reach marginalized populations at increased risk for HIV acquisition. These strategies address not only regional differences, but also the unique cultural nuances of these diverse populations. These nuances included respectfully meeting people where they were and in ways with which they were already familiar, without expecting them to change their behavior in order to learn about the study. This meant, for example, being present in community locales for face-to-face outreach, and advertising on internet dating websites. The consultations with the community to develop the lists of recruitment strategies for each region and the identification of common time points to best compare these strategies across clinical research sites were central to this effort.

Future research will need to further explore more precise methods of data capture, as well as the true costs associated with implementing multifaceted recruitment strategies. While our project points to fiscal implications associated with various recruitment tactics, a more detailed financial analysis would be useful to inform the field. Additionally, any analysis of cost effectiveness will need to consider the costs of human time and effort, and the use of strategies that may be less expensive but not necessarily more effective. These considerations can vary by site, by community, and by region, and warrant a detailed exploration.

With the recent rapid advances in biomedical HIV prevention, further innovations in community engagement will be needed for clinical trials of new HIV prevention methods. Continued sustainable collaboration between researchers and community representatives is needed to proactively develop and adopt standardized metrics to implement and evaluate methods of community engagement.

Additionally, research sites need to capitalize on their knowledge of and expertise with the local community to tailor strategies for their local environment, and be able to evaluate their own site data as a means of monitoring their performance and improving their efficiency. Sites and communities are not homogenous, and funders need to consider variations in local customization.

## Acknowledgments

The authors thank the 43 research sites that participated in this effort and provided screening and enrollment data for analysis, and the Community Engagement staff and CAB members who served on the respective study Community Working Groups and provided input. We also thank the volunteers in both AMP studies for their contributions to these landmark trials. We are grateful to the leadership and members of the AMP study Protocol Teams for their unwavering support of community engagement efforts in these studies.

## Author Contributions

**Conceptualization:** Gail B. Broder, Stephaun E. Wallace, Michele P. Andrasik.

**Data curation:** Gail B. Broder, Jonathan P. Lucas.

**Formal analysis:** Gail B. Broder, Jonathan P. Lucas, Michele P. Andrasik.

**Investigation:** Gail B. Broder, Jonathan P. Lucas.

**Methodology:** Gail B. Broder, Jonathan P. Lucas, Michele P. Andrasik.

**Project administration:** Gail B. Broder, Jonathan P. Lucas, Jontraye Davis, Stephaun E. Wallace, Nandisile Luthuli, Kagisho Baepanye, Rhonda R. White, Marcus Bolton, Cheryl Blanchette.

**Supervision:** Michele P. Andrasik.

**Writing – original draft:** Gail B. Broder, Jonathan P. Lucas.

**Writing – review & editing:** Gail B. Broder, Jonathan P. Lucas, Jontraye Davis, Stephaun E. Wallace, Nandisile Luthuli, Kagisho Baepanye, Rhonda R. White, Marcus Bolton, Cheryl Blanchette, Michele P. Andrasik.

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
