## [Decision Letter · Decision Letter 0]

6 May 2020

PONE-D-20-04045

Standardized Metrics Can Reveal Region-Specific Opportunities in Community Engagement to Aid Recruitment in HIV Prevention Trials

PLOS ONE

Dear Ms. Broder,

Thank you for submitting your manuscript to PLOS ONE. After careful consideration, we feel that it has merit but does not fully meet PLOS ONE’s publication criteria as it currently stands. Therefore, we invite you to submit a revised version of the manuscript that addresses the points raised during the review process.

We would appreciate receiving your revised manuscript by Jun 20 2020 11:59PM. To enhance the reproducibility of your results, we recommend that if applicable you deposit your laboratory protocols in protocols.io, where a protocol can be assigned its own identifier (DOI) such that it can be cited independently in the future. For instructions see: http://journals.plos.org/plosone/s/submission-guidelines#loc-laboratory-protocols

We look forward to receiving your revised manuscript.

Kind regards,

Jose A. Bauermeister, MPH, PhD

Academic Editor

PLOS ONE

Journal Requirements:

2. Thank you for stating the following beneath the Acknowledgments Section of your manuscript:

'Funding

The HVTN 703/HPTN 081 and HVTN 704/HPTN 085 studies are supported by National Institute of Allergy and Infectious Diseases (NIAID) U.S. Public Health Service grants:

 UM1 AI068614 (Leadership and Operations Center: HIV Vaccine Trials Network),

 UM1 AI068618 (Laboratory 310 Center: HIV Vaccine Trials Network),

 UM1 AI068635 (Statistical and Data Management Center: HIV Vaccine Trials Network).

 UM1 AI068619 (Leadership and Operations Center: HIV Prevention Trials Network)

 UM1 AI068613 (Laboratory Center: HIV Prevention Trials Network)

 UM1 AI068617 (Statistical and Data Management Center: HIV Prevention Trials Network)

The content is solely the responsibility of the authors and does not necessarily represent the official views of the NIAID or NIH.'

'The authors received no specific funding for this work.'

3. We noted in your submission details that a portion of your manuscript may have been presented or published elsewhere:

'Portions of this manuscript were presented as an oral abstract at the HIVR4P conference in Madrid, Spain in October, 2018. The abstract from the conference presentation was published in the conference program and made available on their website. The manuscript expands on this presentation substantively. '

Please clarify whether this conference proceeding was peer-reviewed and formally published. If this work was previously peer-reviewed and published, in the cover letter please provide the reason that this work does not constitute dual publication and should be included in the current manuscript.

4. One of the noted authors is a group, 'HVTN 703/HPTN 081 and HVTN 704/HPTN 085 Protocol Teams'.

In addition to naming the author group, please list the individual authors and affiliations within this group in the acknowledgments section of your manuscript.

Please also indicate clearly a lead author for this group along with a contact email address.

Reviewers' comments:

Reviewer's Responses to Questions

**Comments to the Author**

1. Is the manuscript technically sound, and do the data support the conclusions?

Reviewer #1: Yes

Reviewer #2: Yes

2. Has the statistical analysis been performed appropriately and rigorously? 

Reviewer #1: Yes

Reviewer #2: Yes

3. Have the authors made all data underlying the findings in their manuscript fully available?

Reviewer #1: Yes

Reviewer #2: Yes

4. Is the manuscript presented in an intelligible fashion and written in standard English?

Reviewer #1: Yes

Reviewer #2: Yes

5. Review Comments to the Author

Reviewer #1: This study aims to create a metric to describe, monitor, and evaluate good participatory practice (GPP) guidelines in HIV monoclonal antibody trials. Overall, this manuscript is well-written and organized. Major comments apply to the intro and methods sections.

Introduction:

In the Introduction it reads like the study’s effectiveness at adhering to the principles of respect, mutual understanding, integrity, etc…. will be evaluated in this study. It isn’t until the very end that it becomes evident that the purpose of this study is to assess the efficiency of the recruitment strategies. This should be made explicit earlier on and more clearly worked into the logic of the study rationale in the introduction.

Methods:

It is not very evident that this paper is basically evaluating the effectiveness of different recruitment strategies in different settings. The focus on the GPP guidelines takes away from this. Authors should consider focusing on how these guidelines apply to recruitment practices and how recruitment using GPP guidelines differs from other types of recruitment. It is really not clear why GPP guidelines are important when the meat of this study is basically in the answer to the question, “how did you hear about this study?”.

Reviewer #2: This is a clearly written manuscript that uses data on recruitment practices and efficiencies across multiple sites of two clinical trials to identify preferred strategies and enrolment ratios. The multiple sites and data points is a strength of the manuscript. However, there are some potential discrepancies in characterizations of the approach and its implementation, and interpretations of the results. Finally, the manuscript would benefit significantly from delving into the rich field of research, including empirical research as well as the few descriptive studies cited (none in the discussion), to characterize the many tensions and challenges that underlie what is presented as a rather neutral and uncomplicated process of community engagement (CE). In addition, there are many studies of recruitment practices. Engaging with the existing literature would make the findings and insights gained much more valuable and serve to better advance the field. There are far too many acronyms, particularly for a general scientific audience that reads PLOS One: suggest using some but limiting substantially.

Introduction:

It would be helpful to include and discuss relevant literature on community engagement in, particularly, HIV prevention trials, much of which is omitted. At least these need to be referenced and addressed in the discussion.

Line 72: what is a “bnAb”?

Line 77: break down n=2701 into n’s for cisgender and transgender men, and transgender women. (Also, one wonders if there are differentially effective CE and recruitment methods for different subpopulations?)

Line 86: broader CE than? Usual HVTN/HPTN practices? Or?

Line 89: this is surely a substantial list of principles. However, it is unclear how each of these were addressed in your analysis. I do not recall mention of or disaggregation of these principles in the results or discussion.

Also, what principles were omitted? And why?

Lines 90-96: While the GPP are certainly widely discussed guidelines, they are not without challenges; and a number of articles have evoked or discussed these. This needs to be acknowledged here, or in the discussion section. See list at end

Line 99: Too many acronyms. Are the CRSs considered local? Or beyond? What about NCEP staff? The question arises as to how many local staff are permanent; and how many international staff are permanent.

Line 100: why does the CWG only include one CAB rep and one community educator? Some studies have identified challenges in local community staff raising concerns or speaking frankly with international staff and investigators, more so as the latter pay the salaries of local staff. See references below. Arguably this may not be such a fraught issue in discussing study recruitment; however this should be acknowledged in the discussion.

METHODS

There seems to be a discrepancy between the statement on 119 (“underlying assumption that community members have the best insight regarding which recruitment strategies work best in their local cultural and geographic context”, and line 131, “NCEP staff for HVTN and HPTN developed a list of recruitment strategies based on their experiences working with the sites” and shared the lists with CWG members who “revised them for accuracy.” This may reflect a need to better describe the process; but why did the initial lists not emerge from “community members” if they are the experts? Might having lists pre-arranged by HVTN and HPTN staff influence the listing and choices of community members?

Lines 135-139: Here is an interesting example of different approaches taken by African CWG and Americas and Swiss CWG to characterize referral sources. Since you are advocating “common metrics”, how might these differences influence the metric devised? Should there be flexibility underlying the metrics and their application to enable different sites/site groups to apply them more accurately? How far does the flexibility go?

Lines 152-157: It would be informative to disaggregate numbers “found ineligible to enroll” and those who were “eligible but decided not to participate”? Those numbers are treated differently in clinical trials; and they could have different ramifications for recruitment practices: i.e., targeting the ‘wrong’ population vs. challenges in willingness to participate.

RESULTS

Later you do mention the need to include cost effectiveness analysis in future studies, which is fine. However, you might specifically mention that raw numbers of screening to enrollment ratios per recruitment strategy might not necessarily support an evidence informed decision to prioritize one over the other. For ex., if “transit advertising” were relatively much less expensive than face-to-face outreach, arguably the former might be an acceptable strategy, or some combination thereof.

DISCUSSION

The complete lack of references and discussion of the present results in the context of many other studies of CE in HIV prevention trials and global health more broadly, as well as studies of recruitment, is a shortcoming and a lost opportunity to engage these findings with the existing literature. For ex., what are the challenges of “consulting with the community to identify recruitment strategies most appropriate for each region”? Who represents “the community”? Who decides on the representatives? From whom do the community representatives and community staff draw salaries? Are they permanent or temporary staff? What happens if different representatives do not agree? Arguably pastors and gay and transgender people do not likely agree on many important issues, which may impact on willingness of the latter to speak openly; and the recruitment strategies and target communities identified. See references below.

While this study does present helpful ideas and data on using community processes to identify metrics to identify preferred and effective recruitment strategies, what are some of the challenges in harmonizing metrics amidst cultural, geographical, and economic differences? In the context of stigma, criminalization of PLHIV and at-risk communities in some sites? Vast differences in wealth and power between trial sponsor nations and LMIC in which many trials are conducted? This is not to lessen the importance or potential utility of the findings; but it would make them more interesting and applicable in the real world if some of the most relevant literature and challenges were acknowledged, with discussion of how the development of such metrics might address or adapt to challenges in the face of disparate cultures, geographies, wealth and power that underlie the realities of global health research and HIV prevention and other clinical trials. See below:

CE References, many of which are highly relevant:

de Wet, A, Swartz, L, Kagee, A, et al. (2019): The trouble with difference: Challenging and reproducing inequality in a biomedical HIV research community engagement process, Global Public Health, DOI: 10.1080/17441692.2019.1639209

Kagee, A., De Wet, A., Kafaar, Z. et ak, (2020). Caveats and pitfalls associated with researching community engagement in the context of HIV vaccine trials. Journal of Health Psychology, 25(1), 82–91. https://doi.org/10.1177/1359105317745367

Lavery, James V. et al. Towards a framework for community engagement in global health research. Trends in Parasitology, Volume 26, Issue 6, 279 – 283. DOI: https://doi.org/10.1016/j.pt.2010.02.009

Molyneux, S., Sariola, S., Allman, D., et al. (2016). Public/community engagement in health research with men who have sex with men in sub-Saharan Africa: Challenges and opportunities. Health Research Policy and Systems, 14, 65.

Montgomery, C. M., Sariola, S., Kingori, P., & Engel, N. (2017). Global health and science and technology studies: Complacency and critique. Science and Technology Studies, 30(4), 1–13

Newman PA, Rubincam C, Slack C,et al. (2015) Towards a Science of Community Stakeholder Engagement in Biomedical HIV Prevention Trials: An Embedded Four-Country Case Study. PLoS ONE 10(8): e0135937. doi:10.1371/journal.pone.0135937

Newman, PA & Rubincam, C. (2014) Advancing community stakeholder engagement in biomedical HIV prevention trials: principles, practices and evidence, Expert Review of Vaccines, 13:12, 1553-1562, DOI: 10.1586/14760584.2014.953484

Rubincam, C., et al. (2015): Taking culture seriously in biomedical HIV prevention trials: a meta-synthesis of qualitative studies, Expert Review of Vaccines, DOI: 10.1586/14760584.2016.1118349

Thabethe, S., Slack, C., (2018). “Why Don’t You Go Into Suburbs? Why Are You Targeting Us?”: Trust and Mistrust in HIV Vaccine Trials in South Africa. Journal of Empirical Research on Human Research Ethics, 13(5), 525–536. https://doi.org/10.1177/1556264618804740

Recruitment (among many others)

Horvath KJ, Nygaard K, Danilenko GP, et al. Strategies to retain participants in a long-term HIV prevention randomized controlled trial: lessons from the MINTS-II study. AIDS Behav. 2012;16(2):469‐479. doi:10.1007/s10461-011-9957-3

Newman, PA, Duan, N, Roberts, K, et al. HIV Vaccine Trial Participation Among Ethnic Minority Communities: Barriers, Motivators, and Implications for Recruitment, JAIDS Journal of Acquired Immune Deficiency Syndromes: February 1st, 2006 - Volume 41 - Issue 2 - p 210-217 doi: 10.1097/01.qai.0000179454.93443.60

Screening, recruiting and predicting retention of participants in the NIMH Multisite HIV Prevention Trial, AIDS, Supplement: February 1997 - Volume 11 - Issue - p S13-S19

6. PLOS authors have the option to publish the peer review history of their article (what does this mean?). If published, this will include your full peer review and any attached files.

Reviewer #1: No

Reviewer #2: No

---

## [Author Response · Author response to Decision Letter 0]

22 Jul 2020

We thank the reviewers for the detailed comments they provided. We have uploaded a response letter that includes a point by point reply to each of their suggestions.

---

## [Decision Letter · Decision Letter 1]

3 Sep 2020

Standardized Metrics Can Reveal Region-Specific Opportunities in Community Engagement to Aid Recruitment in HIV Prevention Trials

PONE-D-20-04045R1

Dear Dr. Broder,

We’re pleased to inform you that your manuscript has been judged scientifically suitable for publication and will be formally accepted for publication once it meets all outstanding technical requirements.

Kind regards,

Douglas S. Krakower, MD

Academic Editor

PLOS ONE

Additional Editor Comments (optional):

Reviewers' comments:

Reviewer's Responses to Questions

**Comments to the Author**

1. If the authors have adequately addressed your comments raised in a previous round of review and you feel that this manuscript is now acceptable for publication, you may indicate that here to bypass the “Comments to the Author” section, enter your conflict of interest statement in the “Confidential to Editor” section, and submit your "Accept" recommendation.

Reviewer #2: All comments have been addressed

2. Is the manuscript technically sound, and do the data support the conclusions?

Reviewer #2: Yes

3. Has the statistical analysis been performed appropriately and rigorously? 

Reviewer #2: Yes

4. Have the authors made all data underlying the findings in their manuscript fully available?

Reviewer #2: Yes

5. Is the manuscript presented in an intelligible fashion and written in standard English?

Reviewer #2: Yes

6. Review Comments to the Author

Reviewer #2: (No Response)

7. PLOS authors have the option to publish the peer review history of their article (what does this mean?). If published, this will include your full peer review and any attached files.

Reviewer #2: No

---

## [Editor Report · Acceptance letter]

7 Sep 2020

PONE-D-20-04045R1 

Standardized Metrics Can Reveal Region-Specific Opportunities in Community Engagement to Aid Recruitment in HIV Prevention Trials 

Dear Dr. Broder:

I'm pleased to inform you that your manuscript has been deemed suitable for publication in PLOS ONE. Congratulations! Your manuscript is now with our production department. 

Kind regards, 

on behalf of

Dr. Douglas S. Krakower 

Academic Editor

PLOS ONE